# Inferring neural population dynamics from multiple partial recordings of the same neural circuit

Srinivas C. Turaga[*1,2], Lars Buesing[1], Adam M. Packer[2], Henry Dalgleish[2], Noah Pettit[2], Michael Häusser[2] and Jakob H. Macke[3,4]

[1]Gatsby Computational Neuroscience Unit, University College London
[2]Wolfson Institute for Biomedical Research, University College London
[3]Max-Planck Institute for Biological Cybernetics, Tübingen
[4]Bernstein Center for Computational Neuroscience, Tübingen

## Abstract

Simultaneous recordings of the activity of large neural populations are extremely valuable as they can be used to infer the dynamics and interactions of neurons in a local circuit, shedding light on the computations performed. It is now possible to measure the activity of hundreds of neurons using 2-photon calcium imaging. However, many computations are thought to involve circuits consisting of thousands of neurons, such as cortical barrels in rodent somatosensory cortex. Here we contribute a statistical method for "stitching" together sequentially imaged sets of neurons into one model by phrasing the problem as fitting a latent dynamical system with missing observations. This method allows us to substantially expand the population-sizes for which population dynamics can be characterized—beyond the number of simultaneously imaged neurons. In particular, we demonstrate using recordings in mouse somatosensory cortex that this method makes it possible to predict noise correlations between non-simultaneously recorded neuron pairs.

## 1 Introduction

The computation performed by a neural circuit is a product of the properties of single neurons in the circuit and their connectivity. Simultaneous measurements of the collective dynamics of all neurons in a neural circuit will help us understand their function and test theories of neural computation. However, experimental limitations make it difficult to measure the joint activity of large populations of neurons. Recent progress in 2-photon calcium imaging now allows for recording of the activity of hundreds of neurons nearly simultaneously [1, 2]. However, in neocortex where circuits or subnetworks can span thousands of neurons, current imaging techniques are still inadequate.

We present a computational method to more effectively leverage currently available experimental technology. To illustrate our method consider the following example: A whisker barrel in the mouse somatosensory cortex consists of a few thousand neurons responding to stimuli from one whisker. Modern microscopes can only image a small fraction—a few hundred neurons—of this circuit. But since nearby neurons couple strongly to one another [3], by moving the microscope to nearby locations, one can expect to image neurons which are directly coupled to the first population of neurons. In this paper we address the following question: Could we characterize the joint dynamics of the first and second populations of neurons, even though they were not imaged simultaneously? Can we estimate correlations in variability across the two populations? Surprisingly, the answer is yes.

We propose a statistical tool for "stitching" together measurements from multiple partial observations of the same neural circuit. We show that we can predict the correlated dynamics of large

---

[*]sturaga@gatsby.ucl.ac.uk

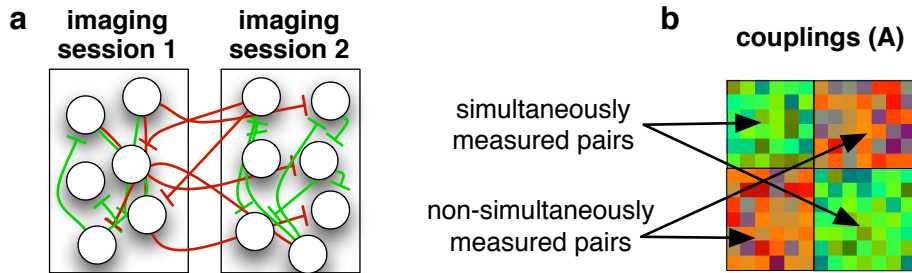

Figure 1: **Inferring neuronal interactions from non-simultaneous measurements.** **a)** If two subsets of a neural population can only be recorded from in two separate imaging sessions, can we infer the connectivity across the sub-populations (red connections)? **b)** We want to infer the functional connectivity matrix, and in particular those entries which correspond to pairs of neurons that were not simultaneously measured (red off-diagonal block). While the two sets of neurons are pictured as non-overlapping here, we will also be interested in the case of partially overlapping measurements.

populations of neurons even if many of the neurons have not been imaged simultaneously. In sensory cortical neurons, where large variability in the evoked response is observed [4, 5], our model can successfully predict the magnitude of (so-called) noise correlations between non-simultaneously recorded neurons. Our method can help us build data-driven models of large cortical circuits and help test theories of circuit function.

**Related recent research.** Numerous studies have addressed the question of inferring functional connectivity from 2-photon imaging data [6, 7] or electrophysiological measurements [8, 9, 10, 11]. These approaches include detailed models of the relationship between fluorescence measurements, calcium transients and spiking activity [6] as well as model-free information-theoretic approaches [7]. However, these studies do not attempt to infer functional connections between non-simultaneously observed neurons. On the other hand, a few studies have presented statistical methods for dealing with sub-sampled observations of neural activity or connectivity, but these approaches are not applicable to our problem: A recent study [12] presented a method for predicting noise correlations between non-simultaneously recorded neurons, but this method requires the strong assumption that noise correlations are monotonically related to stimulus correlations. [13] presented an algorithm for latent GLMs, but this algorithm does not scale to the population sizes of interest here. [14] presented a method for inferring synaptic connections on dendritic trees from sub-sampled voltage observations. In this setting, one typically obtains a measurement from each location every few imaging frames, and it is therefore possible to interpolate these observations. In contrast, in our application, imaging sessions are of much longer duration than the time-scale of neural dynamics. Finally, [15] presented a statistical framework for reconstructing anatomical connectivity by superimposing partial connectivity matrices derived from fluorescent markers.

## 2 Methods

Our goal is to estimate a joint model of the activity of a neural population which captures the correlation structure and stimulus selectivity of the population from partial observations of the population activity. We model the problem as fitting a latent dynamical system with missing observations. In principle, any latent dynamical system model [13] can be used—here we demonstrate our main point using the simple linear gaussian dynamical system for its computational tractability.

### 2.1 A latent dynamical system model for combining multiple measurements of population activity

**Linear dynamics.** We denote by $x^k$ the activity of $N$ neurons in the population on recording session $k$, and model its dynamics as linear with Gaussian innovations in discrete time,

$$x_t^k = Ax_{t-1}^k + Bu_t^k + \eta_t, \qquad\qquad \text{where } \eta_t \sim \mathcal{N}(0, Q). \qquad (1)$$

Here, the $N \times N$ coupling matrix $A$ models correlations across neurons and time. An entry $A_{ij}$ being non-zero implies that activity of neuron $j$ at time $t$ has a statistical influence on the activity of neuron $i$ on the next time-step $t+1$, but does not necessarily imply a direct synaptic connection. For this reason, entries of $A$ are usually referred to as the 'functional' (rather than anatomical) couplings or connectivity of the population. The entries of $A$ also shape trial-to-trial variability which is correlated across neurons, i.e. noise-correlations. Further, we include an external, observed stimulus $u_t^k$ (of dimension $N_u$) as well as receptive fields $B$ (of size $N \times N_u$) which model the stimulus dependence of the population activity. We model neural noise (which could include the effect of other influences not modeled explicitly) using zero-mean innovations $\eta_t$, which are Gaussian i.i.d. with covariance matrix $Q$, assuming the latter to be diagonal (see below for how our framework also can allow for correlated noise). The mean $x_0$ and covariance $Q_0$ of the initial state $x_0^k$ were chosen such that the system is stationary (apart from the stimulus contribution $Bu_t^k$), i.e. $x_0 = 0$ and $Q_0$ satisfies the Lyapunov equation $Q_0 = AQ_0A^\top + Q$.

For the sake of simplicity, we work directly in the space of continuous valued imaging measurements (rather than on the underlying spiking activity), i.e. $x_t^k$ models the relative calcium fluorescence signal. While this model does not capture the nonlinear and non-Gaussian cascade of neural couplings, calcium dynamics, fluorescence measurements and imaging noise [16, 6], we will show that this model nevertheless is able to predict correlations across non-simultaneously observed pairs of neurons.

**Incomplete observations.** In each imaging session $k$ we measure the activity of $N_k$ neurons simultaneously, where $N_k$ is smaller than the total number of neurons $N$. Since these measurements are noisy and incomplete observations of the full state vector, the true underlying activity of all neurons $x_t^k$ is treated as a latent variable. The vector of the $N_k$ measurements at time $t$ in session $k$ is denoted as $y_t^k$ and is related to the underlying population activity by

$$y_t^k = C^k(x_t^k + d + \epsilon_t) \qquad\qquad \epsilon_t \sim \mathcal{N}(0, R), \qquad (2)$$

where the 'measurement matrix' $C^k$ is of size $N_k \times N$. Further assuming that the recording sites correspond to identified cells (which typically is the case for 2-photon calcium imaging), we can assume $C^k$ to be known and of the following form: The element $C_{ij}^k$ is 1 if neuron $j$ of the population is being recorded from on session $k$ (as the $i$-th recording site); the remaining elements of $C^k$ are 0. The measurement noise is modeled as a Gaussian random variable $\epsilon_t$ with covariance $R$, and the parameter $d$ captures a constant offset. One can also envisage using our model with dimensions of $x_t^k$ which are *never* observed– such latent dimensions would then model correlated noise or the input from unobserved neurons into the population [17, 18].

**Fitting the model.** Our goal is to estimate the parameters $(A, B, Q, R)$ of the latent linear dynamical system (LDS) model described by equations (1) and (2) from experimental data. One can learn these parameters using the standard expectation maximization (EM) algorithm that finds a local maximum of the log-likelihood of the observed data [19]. The E-step can be performed via Kalman Smoothing (with a different $C^k$ for each session). In the M-step, the updates for $A$, $B$ and $Q$ are as in standard linear dynamical systems, and the updates for $R$ and $d$ are element-wise given by

$$d_j = \frac{1}{Tn_j} \sum_{k,t} \chi_j^k \left( y_{t,\sigma_j^k}^k - \left\langle x_{t,j}^k \right\rangle \right)$$

$$R_{jj} = \frac{1}{Tn_j} \sum_{k,t} \chi_j^k \left\langle (y_{t,\sigma_j^k}^k - x_{t,j}^k - d_j)^2 \right\rangle,$$

where $\langle \cdot \rangle$ denotes the expectation over the posterior distribution calculated in the E-step, and $T$ is the number of time steps in each recording session (assumed to be the same for each session for the sake of simplicity). Furthermore, $\chi_j^k := \sum_i C_{ij}^k$ is 1 if neuron $j$ was imaged in session $k$ and 0 otherwise, $n_j = \sum_k \chi_j^k$ is the total number of sessions in which neuron $j$ was imaged and $\sigma_j^k$ is the index of the recording site of neuron $j$ during session $k$. To improve the computational efficiency of the fitting procedure as well as to avoid shallow local maxima, we used a variant of online-EM with randomly selected mini-batches [20] followed by full batch EM for fine-tuning.

## 2.2 Details of simulated and experimental data

**Simulated data.** We simulated a population of 60 neurons which were split into 3 pools ('cell types') of 20 neurons each, with both connection probability and strength being cell-type specific. Within each pool, pairs were coupled with probability $50\%$ and random weights, cell-types one and two had excitatory connections onto the other cells, and type three had weak but dense inhibitory couplings (see Figure 2a, top left). Coupling weights were truncated at $\pm 0.2$. The 4-dimensional external stimulus was delivered into the first pool. On average, $24\%$ of the variance of each neuron was noise, $2\%$ driven by the stimulus, $25\%$ by self-couplings and a further $49\%$ by network-interactions. After shuffling the ordering of neurons (resulting in the connectivity matrix displayed in Fig. 2a, top middle), we simulated $K = 10$ trials of length $T = 1000$ samples from the population. We then pretended that the population was imaged in two sessions with non-overlapping subsets of 30 neurons each (Figure 2a, green outlined blocks) of $K = 5$ trials each, and that observation noise $\epsilon$ was uncorrelated and very small, $\text{std}(\epsilon_{ii}) = 0.006$.

**Experimental data.** We also applied the stitching method to two calcium imaging datasets recorded in the somatosensory cortex of *awake* or *anesthetized* mice. We imaged calcium signals in the superficial layers of mouse barrel cortex (S1) *in-vivo* using 2-photon laser scanning microscopy [1]. A genetically encoded calcium indicator (GCaMP6s) was virally expressed, driven pan-neuronally by the human-synapsin promoter, in the C2 whisker barrel and the activity of about 100-200 neurons was imaged simultaneously in-vivo at about 3Hz, compatible with the slow timescales of the calcium dynamics revealed by GCaMP6s. The *anesthetized* dataset was collected during an experiment in which the C2 whisker of an anesthetized mouse was repeatedly flicked randomly in one of three different directions (rostrally, caudally or ventrally). About 200 neurons were imaged for about 27min at a depth of $240\mu$m in the C2 whisker barrel. The *awake* dataset was collected while an awake animal was performing a whisker flick detection task. In this session, about 80 neurons were imaged for about 55min at a depth of $190\mu$m, also in the C2 whisker barrel. Regions of interest (ROI) corresponding to putative GCaMP expressing soma (and in some instances isolated neuropil) were manually defined and the time-series corresponding to the calcium signal for each such ROI was extracted. The calcium time-series were high-pass filtered with a time-constant of 1s.

## 2.3 Quantifying and comparing model performance

**Fictional imaging scenario in experimental data.** To evaluate how well stitching works on real data, we created a fictional imaging scenario. We pretended that the neurons, which were in reality simultaneously imaged, were not imaged in one session but instead were 'imaged' in two subsets in two different sessions. The subsets corresponding to different 'sessions' $c = 60\%$ of the neurons, meaning that the subsets overlapped and a few neurons in common. We also experimented with $c = 50\%$ as in our simulation above, but failed to get good performance without any overlapping neurons. We imagined that we spent the first $40\%$ of the time 'imaging' subset 1 and the second $40\%$ of the time 'imaging' subset 2. The final $20\%$ of the data was withheld for use as the test set. We then used our stitching method to predict pairwise correlations from the fictional imaging session.

**Upper and lower bounds on performance.** We wanted to benchmark how well our method is doing both compared to the theoretical optimum and to a conventional approach. On synthetic data, we can use the ground-truth parameters as the optimal model. In lieu of ground-truth on the real data, we fit a 'fully observed' model to the simulatenous imaging data of all neurons (which would be impossible of course in practice, but is possible in our fictional imaging scenario). We also analyzed the data using a conventional, 'naive' approach in which we separately fit dynamical system models to each of the two imaging sessions and then combined their parameters. We set coefficients of non-simultaneously recorded pairs to 0 and averaged coefficients for neurons which were part of both imaging sessions (in the $c = 60\%$ scenario). The "fully observed" and the "naive" models constitute an upper and lower bound respectively on our performance. Certainly we can not expect to do better at predicting correlations, than if we had observed all neurons simultaneously.

## 3 Results

We tested our ability to stitch multiple observations into one coherent model which is capable of predicting statistics of the joint dynamics, such as correlations across non-simultaneously imaged

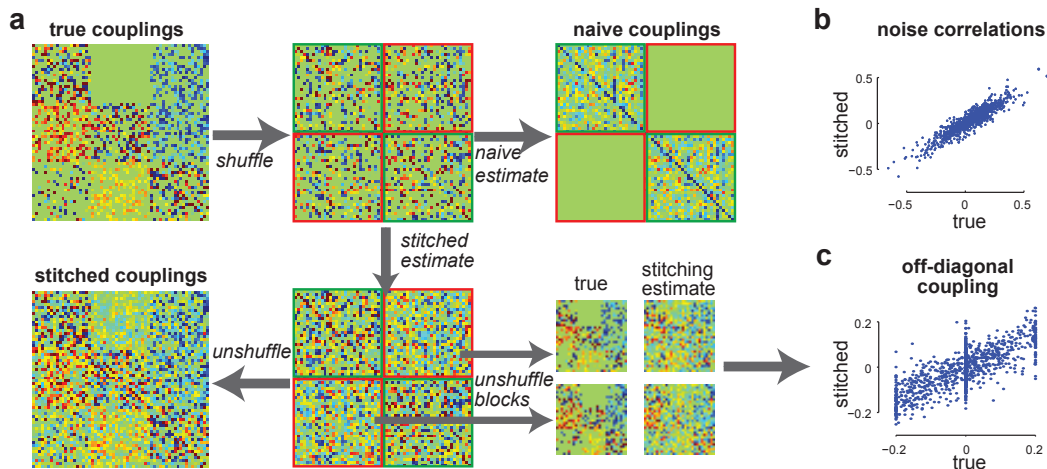

Figure 2: **Noise correlations and coupling parameters can be well recovered in a simulated dataset. a)** A coupling matrix for 60 neurons arranged in 3 blocks was generated (true coupling matrix) and shuffled. We simulated the imaging of non-overlapping subsets of 30 neurons each in two sessions. Couplings were recovered using a "naive" strategy and using our proposed "stitching" method. **b)** Noise correlations estimated by our stitching method match true noise correlations well. **c)** Couplings between non-simultaneously imaged neuron pairs (red off-diagonal block) are estimated well by our method.

neuron pairs. We first apply our method to a synthetic dataset to explain its properties, and then demonstrate that it works for real calcium imaging measurements from the mouse somatosensory cortex.

## 3.1 Inferring correlations and model parameters in a simulated population

It might seem counterintuitive that one can infer the cross-couplings, and hence noise-correlations, between neurons observed in separate sessions. An intuition for why this might work nevertheless can gained by considering the artificial scenario of a network of linearly interacting neurons driven by Gaussian noise: Suppose that during the first recording session we image half of these neurons. We can fit a linear state-space model to the data in which the other, unobserved half of the population constitutes the latent space. Given enough data, the maximum likelihood estimate of the model parameters (which is consistent) lets us identify the true joint dynamics of the whole population up to an invertible linear transformation of the unobserved dimensions [21]. After the second imaging session, where we image the second (and previously unobserved) half of the population, we can identify this linear transformation, and thus identify all model parameters uniquely, in particular the cross-couplings. To demonstrate this intuition, we simulated such an artificial dataset (described in 2.2) and describe here the results of the stitching procedure.

**Recovering the coupling matrix.** Our stitching method was able to recover the true coupling matrix, including the off-diagonal blocks which correspond to pairs of neurons that were not imaged simultaneously (see red-outlined blocks in 2a, bottom middle). As expected, recovery was better for couplings across observed pairs (correlation between true and estimated parameters $0.95$, excluding self-couplings) than for non-simultaneously recorded pairs (Figure 2c; correlation $0.73$). With the "naive" approach couplings between non-simultaneously observed pairs cannot be recovered, and even for simultaneously observed pairs, the estimate of couplings is biased (correlation $0.75$).

**Recovering noise correlations.** We also quantified the degree to which we are able to predict statistics of the joint dynamics of the whole network, in particular noise correlations across pairs of neurons that were never observed simultaneously. We calculated noise correlations by computing correlations in variability of neural activity after subtracting contributions due to the stimulus. We found that the stitching method was able to accurately recover the noise-correlations of non-simultaneously recorded pairs (correlation between predicted and true correlations was $0.92$; Figure 2b). In fact, we generally found the prediction of correlations to be more accurate than prediction

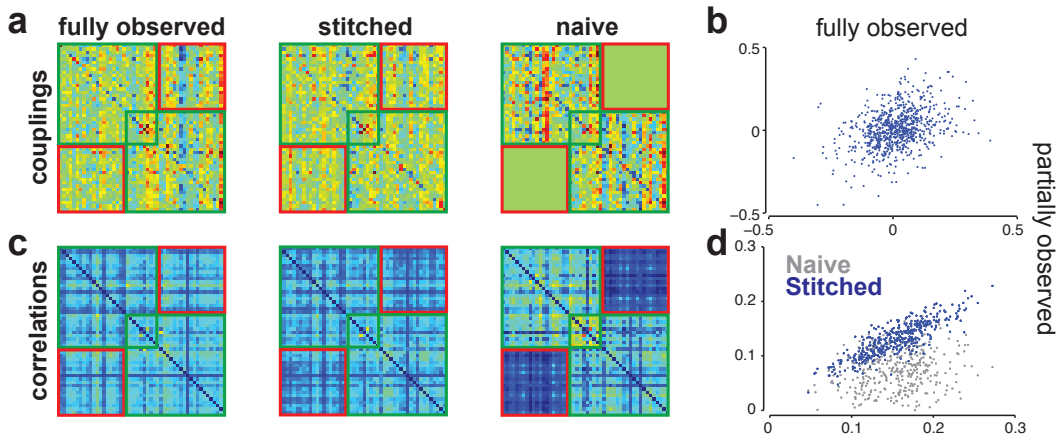

Figure 3: **Examples of correlation and coupling recovery in the anesthetized calcium imaging experiments. a)** Coupling matrices fit to calcium signal using all neurons (fully observed) or fit after "imaging" two overlapping subsets of 60% neurons each (stitched and naive). The naive approach is unable to estimate coupling terms for "non-simultaneously imaged" neurons, so these are set to zero. **b)** Scatter plot of coupling terms for "non-simultaneously imaged" neuron pairs estimated using the stitching method vs the fully observed estimates. **c)** Correlations predicted using the coupling matrices. **d)** Scatter plot of correlations in **c** for "non-simultaneously imaged" neuron pairs estimated using the stitching and the naive approaches.

of the underlying coupling parameters. In contrast, a naive approach would not be able to estimate noise correlations between non-simultaneously observed pairs. (We note that, as the stimulus drive in this simulation was very weak, inferring noise correlations from stimulus correlations [12] would be impossible).

**Predicting unobserved neural activity.** Given activity measurements from a subset of neurons, our method can predict the activity of neurons in the unobserved subset. This prediction can be calculated by doing inference in the resulting LDS, i.e. by calculating the posterior mean $\mu_{1:T}^k = E(x_{1:T}^k|y_{1:T}^k, h_{1:T}^k)$ and looking at those entries of $\mu_{1:T}^k$ which correspond to unobserved neurons. On our simulated data, we found that this prediction was strongly correlated with the underlying ground-truth activity (average correlation $0.70 \pm 0.01$ s.e.m across neurons, using a separate test-set which was not used for parameter fitting.). The upper bound for this prediction metric can be obtained by using the ground-truth parameters to calculate the posterior mean. Use of this ground-truth model resulted in a performance of $0.82 \pm 0.01$. In contrast, the 'naive' approach can only utilize the stimulus, but not the activity of the observed population for prediction and therefore only achieved a correlation of $0.23 \pm 0.01$.

## 3.2 Inferring correlations in mouse somatosensory cortex

Next, we applied our stitching method to two real datasets: *anesthetized* and *awake* (described in Section 2.2). We demonstrate that it can predict correlations between non-simultaneously accessed neuron pairs with accuracy approaching that of the upper bound ("fully observed" model trained on all neurons), and substantially better than the lower bound "naive" model.

**Example results.** Figure 3a displays coupling matrices of a population consisting of the 50 most correlated neurons in the *anesthetized* dataset (see Section 2.2 for details) estimated using all three methods. Our stitching method yielded a coupling matrix with structure similar to the fully observed model (Figure 3a, central panel), even in the off-diagonal blocks which correspond to non-simultaneously recorded pairs. In contrast, the naive method, by definition, is unable to infer couplings for non-simultaneously recorded pairs, and therefore over-estimates the magnitude of observed couplings (Figure 3a, right panel). Even for non-simultaneously recorded pairs, the stitched model predicted couplings which were correlated with the fully observed predictions (Figure 3b, correlation 0.38).

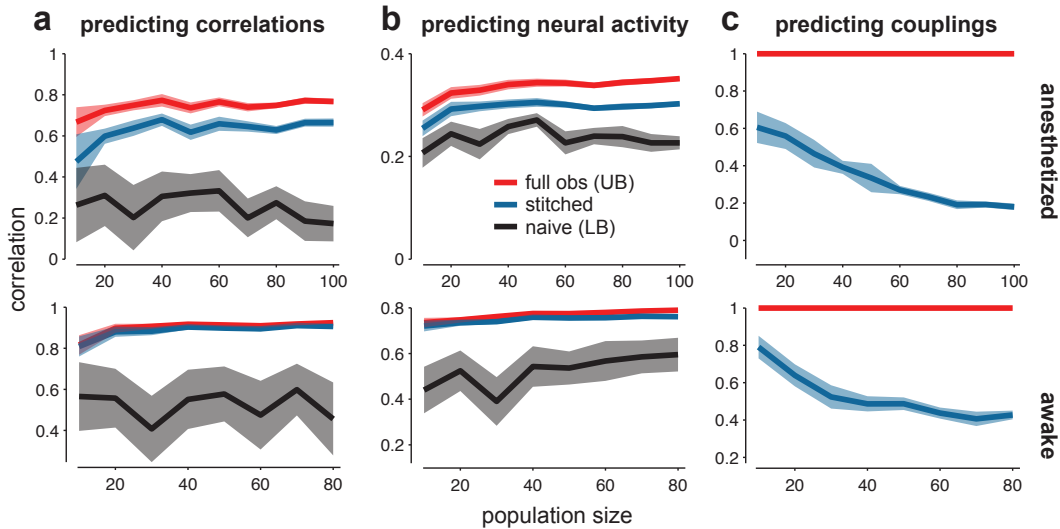

Figure 4: **Recovering correlations and coupling parameters in a real calcium imaging experiments.** 100 neurons were simultaneously imaged in an anesthetized mouse (top row) and an awake mouse (bottom row). Random populations of these neurons, ranging in size from 10 to 100 were chosen and split into two slightly overlapping sub-sets each containing 60% of the neurons. The activity of these sub-sets were imagined to be "imaged" in two separate "imaging" sessions (see Section 2.2). **a)** Pairwise correlations for "non-simultaneously imaged" neuron pairs estimated by the "naive" and our "stitched" strategies compared to correlations predicted by a model fit to all neurons ("full obs"). **b)** Accuracy of predicting the activity of one sub-set of neurons, given the activities of the other sub-set of neurons. **c)** Comparison of estimated couplings for "non-simultaneously imaged" neuron pairs to those estimated using the "fully observed" model. Note that true coupling terms are unavailable here.

However, of greater interest is how well our model can recover pairwise correlations between non-simultaneously measured neuron pairs. We found that our stitching method, but not the naive method, was able to accurately reconstruct these correlations (Figure 3c). As expected, the naive method strongly under-estimated correlations in the non-simultaneously recorded blocks, as it can only model stimulus-correlations but not noise-correlations across neurons. [1] In contrast, our stitching method predicted correlations well, matching those of the fully observed model (correlation 0.84 for stitchLDS, 0.15 for naiveLDS, figure 3d).

**Summary results across multiple populations.** Here, we investigate the robustness of our findings. We drew random neuronal populations of sizes ranging from 10 to 80 (for awake) or 100 (for anesthetized) from the full datasets. For each population, we fit three models (fully observed, stitch, naive) and compared their correlations, parameters and activity cross-prediction accuracy. We repeated this process 20 times for each population size and dataset (anesthetized/awake) to characterize the variability. We found that for both datasets, the correlations predicted by the stitching method for non-simultaneously recorded pairs were similar to the fully observed ones, and that this similarity is almost independent of population size (Figure 4a). In fact, for the awake data (in which the overall level of correlation was higher), the correlation matrices were extremely similar (lower panel). The stitching method also substantially outperformed the naive approach, for which the similarity was lower by a factor of about 2.

We compared the accuracy of the models at predicting the neural activity of one subset of neurons given the stimulus and the activity of the other subset (Figure 4b). We find that our model makes significantly better predictions than the lower bound naive model, whose performance comes from modeling the stimulus and neurons in the overlap between both subsets. Indeed for the more active and correlated *awake* dataset, predictions are nearly as good as those of the fully observed

model. We also found that prediction accuracy increased slightly with population size, perhaps since a larger population provides more neurons from which the activity of the other subset can be predicted. Apparently, this gain in accuracy from additional neurons outweighed any potential drop in performance resulting from increased potential for over-fitting on larger populations.

While we have no access to the true cross-couplings for the real data, we can nonetheless compare the couplings from our stitched model to those estimated by the fully observed model. We find that the stitching model is indeed able to estimate couplings that correlate positively with the fully observed couplings, even for non-simultaneously imaged neuron pairs. Interestingly, this correlation drops with increasing population size, perhaps due to possible near degeneracy of parameters for large systems.

# 4  Discussion

It has long been appreciated that a dynamical system can be reconstructed from observations of only a subset of its variables [22, 23, 21]. These theoretical results suggest that while only measuring the activity of one population of neurons, we can infer the activity of a second neural population that strongly interacts with the first, up to re-parametrization. Here, we go one step further. By later measuring the activity of the second population, we recover the true parametrization allowing us to predict aspects of the joint dynamics of the two populations, such as noise correlations.

Our essential finding is that we can put these theoretical insights to work using a simple linear dynamical system model that "stitches" together data from non-simultaneously recorded but strongly interacting populations of neurons. We applied our method to analyze 2-photon population calcium imaging measurements from the superficial layers of the somatosensory cortex of both anesthetized and awake mice, and found that our method was able to successfully combine data not accessed simultaneously. In particular, this approach allowed us to accurately predict correlations even for pairs of non-simultaneously recorded neurons.

In this paper, we focused our demonstration to stitching together two populations of neurons. Our framework can be generalized to more than two populations, however it remains to be empirically seen how well larger numbers of populations can be combined. An experimental variable of interest is the degree of overlap (shared neurons) between different populations of neurons. We found that some overlap was critical for stitching to work, and increasing overlap improves stitching performance. Given a fixed imaging time budget, determining a good trade-off between overlap and total coverage is an intriguing open problem in experimental design.

We emphasise that our linear gaussian dynamical system provides only a *statistical* description of the observed data. However, even this simple model makes accurate predictions of correlations between non-simultaneously observed neurons. Nevertheless, more realistic models [16, 6] can help improve the accuracy of these predictions and disentangle the contributions of spiking activity, calcium dynamics, fluorescence measurements and imaging noise to the observed statistics. Similarly, better priors on neural connectivity [24] might improve reconstruction performance. Indeed, we found in unreported simulations that using a sparsifying penalty on the connectivity matrix [6] improves parameter estimates slightly. We note that our model can easily be extended to model potential common input from neurons which are never observed [13] as a low dimensional LDS [17, 18].

The simultaneous measurement of the activity of all neurons in a neural circuit will shed much light on the nature of neural computation. While there is much progress in developing faster imaging modalities, there are fundamental physical limits to the number of neurons which can be simultaneously imaged. Our paper suggests a means for expanding our limited capabilities. With more powerful algorithmic tools, we can imagine mapping population dynamics of all the neurons in an entire neural circuit such as the zebrafish larval olfactory bulb, or layers 2 & 3 of a whisker barrel— an ambitious goal which has until now been out of reach.

**Acknowledgements**

We thank Peter Dayan for valuable comments on our manuscript and members of the Gatsby Unit for discussions. We are grateful for support from the Gatsby Charitable Trust, Wellcome Trust, ERC, EMBO, People Programme (Marie Curie Actions) and German Federal Ministry of Education and Research (BMBF; FKZ: 01GQ1002, Bernstein Center Tübingen).

## Footnotes

[1]The naive approach also over-estimated correlations within each view. This is a consequence of biases resulting from averaging couplings across views for neurons in the overlap between the two fictional sessions.

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
