[Reviews · NeurIPS 2013]

Submitted by Assigned_Reviewer_4

The paper contributes a statistical method to “stitch” together sequentially imaged sets of neurons. Such a method expands the population sizes for which population dynamics can be characterized beyond the number of simultaneously imaged neurons and provide a better image of the circuit which cannot be imaged as a whole for experimental and technical limitations. The method is then applied on simulated data as well as on experimental calcium imaging data in mice. Authors show that they can estimate correlations in variability across two populations which were not imaged simultaneously.
The Authors model the problem as fitting a latent dynamical system with missing observations. In this case they used simple linear Gaussian system. Four parameters were defined in the model: a functional coupling matrix, a stimuli receptive fields’ matrix which models the stimulus dependence of the population activity and two covariance matrices for neural and measurement noise. The parameters were estimated from experimental data using a variant of standard EM algorithm.

Quality
The model works nicely on the simulated data with two non-overlapping populations of neurons but when applying on real data populations must have a certain level of overlap to get reasonable results. It is clear why the greater the overlap the better the results are, the question is why wasn’t overlap necessary in the simulated data example?
Also, the authors do not address the source for the great difference in correlation shown when applying the model on simulated data as in fig 2c and on real data as in fig 3b
The beauty of the model is the recovery of pairwise correlations between non-simultaneously measured neuron pairs.

Clarity
The paper is well written, although all figure legends are not descriptive enough and the text related to the figures in the body of the paper does not elaborate enough on the figures.

Originality
The authors list several works which have addressed the question of inferring functional connectivity from 2-photon imaging data or electrophysiological measurements. However, these works do not infer functional connections of non-simultaneously imaged neurons.

Significance
There are many fundamental limits to the number of neurons which can be simultaneously imaged. That of course, limits our capabilities to reveal the nature of the neural computation at the level of the circuit (or more than a few hundreds of cells). This paper suggests a method for expanding these limited capabilities.
Summary: Overall, this is a good model paper, which uses a relatively simple linear dynamics model and captures measures which are not easy to retrieve otherwise.

Submitted by Assigned_Reviewer_6

This paper shows that you can infer network properties of a large network by sequentially recording from subnetworks and then stitching the resulting sub-models. Extended to point-process data, this could be applied to a lot of datasets where the typically a few of the channels are moved every day. I am not really aware of any similar work (J. W. Pillow and P. Latham, “Neural characterization in partially observed populations of spiking neurons” seems to come the closest), and it seems like a powerful idea.

In practice it's not clear how much this method is limited to data that is close to linear-Gaussian, and I don't get a good intuition from the paper how the fitting might break down from various deviations to the assumption. In the extreme case, where the full model can be estimated from one subset (up to a linear transformation of the latent dimensions, as the authors state), it's not clear how much additional information about the coupling matrix is gained from adding on other subsets, as opposed to, say, just using more data from one of the subsets.

Another point that raises some questions is, why does the model on real data only work if there is some overlap between the two population? Obviously this suggests that the linear-Gaussian simulated data does not tell the full story, but it would be nice to have at least some intuition what the overlap does to better constrain the model.

In the experiments on real data, where the linearity assumption probably breaks, one must also wonder how much of the explained noise correlations (which are fit well, the couplings themselves hardly correlate to the true couplings at all in 3b) are due to common input and not direct, pair-wise couplings. I would assume that the noise correlations could be better modeled with latent variables than pairwise couplings.

The authors devote a lot of space to intro and discussion, but are awefully light on the details of the estimation. The equation for the M step on the bottom of page 3 is presented as the main contribution, but not really explained or derived. Maybe it's a trivial result to LDS experts, but that would erode a lot of the novelty of the paper. I think the authors had a very good idea, but it just needs to be explained better to properly convey the significance of the result.

Summary: A neat idea tested against real data, but some details aren't clear.
Author Feedback

Author rebuttal: We thank the reviewers for their thoughtful comments and are pleased that they find it well-written, original and timely, and unanimously recommend our paper for acceptance.

In our paper, we demonstrated a rather simple but consequential idea -- that it is possible to "stitch" the dynamics of large neural populations, while only ever simultaneously recording from a few neurons at a time. In particular, we demonstrate for the first time that it is possible to successfully predict noise-correlations even for non-simultaneously recorded neuron-pairs. Our contribution is an algorithmic approach for overcoming the limitations of current experimental methods for recording neural activity. Our approach based on linear gaussian dynamical systems, while simple, convincingly demonstrates our main claim with surprising accuracy on real data. We performed GCaMP-based 2-photon calcium imaging in layer 2/3 neurons of the mouse somatosensory cortex and showed that we were able to predict correlations between non-simultaneously imaged neurons in both awake and anesthetized conditions.

We thank the reviewers for pointing out ways in which to improve our presentation:
1) As R4 suggests, we will improve our figure legends and descriptions.
2) R5 suggests defining noise correlations more clearly. Indeed, we use the definition of Averbeck et al 2007, as in linear dynamical systems, noise correlations depend both on the system noise and the connectivity matrix.
3) As R6 suggests, we will improve our description of the estimation algorithm.

The reviewers were puzzled by the differences in performance between the toy simulation and the real data. This is because we did not attempt to carefully match the statistics of our simulated network to the real data. Our goal with the simulations was to demonstrate that -- in the best of all worlds -- it is possible to "stitch" together the population model, even with no overlap. Of course we do not expect this idealized scenario to hold, but rather than making our simulations more realistic, we decided to directly test our model on real data. Thus all our main results are based on real calcium imaging datasets.

In line with previous work (Yu et al., 2009; Briggman et al., 2006) and in contrast to our simulations, we find that the neural activity in layer 2/3 neurons are driven largely by low-dimensional common input. This means that only a small subspace of the activity is actually excited by the low-dimensional input, and hence in our model only the corresponding entries of the A-matrix will be constrained to meaningful values. The remaining part of the A-matrix is non-identifiable and unimportant to the neural dynamics. This explains why Fig 3b has a larger scatter than Fig 2c.

Further, we find in Fig 4c that the parameter correlation decreases with increasing population size. This can be interpreted as evidence that the dimensionality of common input (and hence the dimensionality of the constrained parameter space) grows more slowly than the unconstrained noise parameter space. The common input could also be part of the explanation for why some amount of overlap is required for good prediction on the layer 2/3 data.

We agree with the reviewers that improving the simple LDS model to better model common input, the dynamics of the calcium signal and spiking activity will be important directions for future research.